# Assessing perceptions of establishing a vaccine pooled procurement mechanism for the Western Pacific Region

Alice Abou-Nader[1], James D. Heffelfinger[2], Ananda Amarasinghe[2], E. Anthony S. Nelson[1,3]*

**1** Department of Paediatrics, The Chinese University of Hong Kong, Hong Kong, PR China, **2** Vaccine Preventable Diseases and Immunization, World Health Organization Western Pacific Regional Office, Manila, Philippines, **3** School of Medicine, The Chinese University of Hong Kong, Shenzhen, PR China

* tony-nelson@cuhk.edu.hk

**Data Availability Statement:** All relevant data are within the paper and its Supporting Information files.

## Abstract

This study explored the demand and interest among countries in the World Health Organization Western Pacific Region (WPR) to establish and participate in a regional vaccine pooled procurement mechanism. National counterparts affiliated with Ministries of Health that are involved in the national procurement of vaccines within the WPR were identified and invited to complete surveys. Out of 80 counterparts invited, 17 (21%) responded, representing 13 of the 27 WPR countries. Five countries expressed interest in participating in a regional pooled procurement mechanism, 3 expressed lack of interest and 5 did not respond to the question. Preferred characteristics of the procurement mechanism, included flexible participation (i.e. non-compulsory), payment in local currency before receipt of goods and a fixed price for vaccines (i.e. not tiered pricing). Vaccine pricing disparities were noted among upper middle-income and high-income countries for five of the 13 routine vaccines surveyed. Eight countries listed budget planning, quality of vaccines, timely delivery, cost-saving and payment after receipt as potential benefits of pooled procurement.

## Introduction

Procurement mechanisms and vaccine prices help determine whether national governments are able to finance the introduction of vaccines. Since 1979, the Revolving Fund (RF) of the Pan American Health Organizations (PAHO) has compiled the vaccine requirements of the region into a tendering process to create economies of scale, making it a leading example of a pooled procurement mechanism. The RF offers vaccines at fixed and transparent prices to its Member States in the World Health Organization (WHO) Region of the Americas (AMR), regardless of countries' gross national incomes. The fund has simplified the vaccine procurement process and provided more reliable vaccine supplies with fewer vaccine stock-outs [1]. Immunization-specific legislation has been passed in 29 of the 35 countries in the AMR to formalize country participation and financing of vaccines through this regional fund [2, 3]. These developments have supported the expansion of national immunization portfolios, improving

**Funding:** The authors received no specific funding for this work.

**Competing interests:** The authors have declared that no competing interests exist.

access to new and underutilized vaccines. By pooling the vaccine demands in the region, PAHO is able to seek tenders for large vaccine volumes that increase the negotiating power of the secretariat, and consequently reducing the unit prices of vaccines. These fixed and transparent prices enable countries to forecast the vaccine costs for their national immunization programs (NIPs) and make appropriate financial preparations.

Prior to 1991, the 13 Pacific island countries (Cook Islands, Fiji, Kiribati, Marshall Islands, the Federated States of Micronesia, Nauru, Niue, Samoa, Solomon Islands, Tokelau, Tonga, Tuvalu, Vanuatu) were particularly vulnerable to high-priced vaccines as they purchased small volumes and have high transportation costs [4]. In 1991, UNICEF launched the Vaccine Independence Initiative (VII) to address the unique needs of these Pacific island countries. This initiative has allowed countries to have access to credit funds while pool procuring national vaccine requirements through the UNICEF Supply Division (UNCIEF/SD) to ensure quality vaccines were affordable and accessible [4]. Some countries that are eligible to receive financial subsidies for new and under-utilized vaccine introductions from Gavi, the Vaccine Alliance (Gavi) have also been allowed to procure vaccines through the VII without having their national procurement processes interrupted because of insufficient funds [5].

To improve access and supply of vaccines among selected Southeast Asian and Western Pacific countries, the Secretariat of the Association of Southeast Asian Nations (ASEAN) (Brunei Darussalam, Cambodia, Indonesia, Lao People's Democratic Republic, Malaysia, Myanmar, Philippines, Singapore, Thailand, Vietnam) has explored efforts to establish a regional pooled procurement mechanism for vaccines; seven of the 10 ASEAN countries are WHO Western Pacific Region (WPR) Member States (all but Indonesia, Myanmar and Thailand). The ASEAN Vaccine Security and Self-Reliance (AVSSR) initiative was launched in 2014, led by the National Vaccine Institute of Thailand. During 2014–2015, AVSSR conducted workshops in collaboration with National Vaccine Institute of Thailand and WHO and identified vaccine pooled procurement as a means to improve vaccine security and self-reliance among ASEAN Member States and partners [6, 7]. Surveys indicated that an outcome of these AVSSR workshops was support for the establishment of a pooled procurement mechanism for ASEAN Member States [10].

WHO's regional offices drafted tailored strategies to address the 2015 World Health Assembly demand for more transparency on vaccine pricing and similarly the WHO Regional Office for the Western Pacific (WPRO) emphasized the importance of these in its 2015 *Regional Framework for Implementation of the Global Vaccine Action Plan in the Western Pacific* [8]. Exploring pooled negotiation or procurement mechanisms for lower middle-income countries (LMICs) and upper middle-income countries (UMICs) was recommended as a strategy to increase affordability of vaccines in the region [8]. Such a procurement mechanism in the region could reduce vaccine costs for countries in all income categories. Approximately 38% of vaccine manufacturers registered with the Developing Country Vaccine Manufacturer Network are in the WPR [9], highlighting potential opportunities and synergies for these manufacturers to participate in a WPR pooled procurement mechanism. In 2018, the Technical Advisory Group on Immunization and Vaccine-Preventable Diseases in the WPR recommended that the feasibility of a pooled procurement mechanism be explored with the intent of lowering vaccine prices for countries in the region [10].

Building on these efforts, this study explores the perceptions of national stakeholders on the acceptability and feasibility of establishing a regional pooled procurement mechanism for the WHO WPR. Results from this study will help inform decisions on the perceptions of member states and potential cost-savings regarding the establishment of a regional pooled procurement mechanism for vaccines used in national immunization programmes.

## Materials and methods

National stakeholders from ministries of health (MoHs) were invited to complete the survey and were asked to nominate counterparts from ministries of finance (MoFs) and National Immunization Technical Advisory Groups (NITAGs) with which they liaise on vaccine introduction and procurement activities. Tailored surveys were created for MoHs, MoFs, and NITAGs, but all three contained questions addressing four domains: [1] country background information, [2] vaccine pricing, [3] vaccine delivery and quality, [4] perceptions on developing a pooled vaccine procurement mechanism.

### Ethics approval

Ethics approval was granted by the Joint CUHK–New Territories East Cluster Clinical Research Ethics Committee (Ref. CRE-2018.095). The research protocol (Ref. ID.NO.2018.9. HOK.1.EPI) was submitted to the WPRO Ethical Review Committee which determined that it was exempt from review since the intervention was limited to routine evaluation of health programmes.

### Survey design

The four domains of the tailored surveys were based on those previously used in the 2003 study by DeRoeck et al. to assess the possible establishment of a regional pooled procurement mechanism for four newly-independent Central and Eastern European Member States [11]. In this study, the WHO Regional Office for the Europe (EURO) distributed a 37-question survey to national procurement focal points in 15 Central and Eastern European Member States which later informed the topics addressed through subsequent detailed semi-structured interviews of four countries. The responses of the survey and of the guided semi-structured interview questions were assessed to identify recurrent themes that would address our study's objectives (S1 Table and S1 Data). Operational Procedures of the PAHO RF were also referred to when generating survey questions to collect data related to the institutional features that might support a regional pooled procurement mechanism [12]. Follow-up questions were developed based on answers to the previous question and conditional formatting was included within the online survey to avoid any misreporting or inconsistent responses.

### Piloting to improve content of the survey

This study used two pilot phases to gather information and perceptions from potential respondents, identifying unclear language and content. In the first pilot phase, cognitive interviews were conducted with three faculty members from the Department of Paediatrics and School of Public Health of the Chinese University of Hong Kong to assess the layout, clarity, and content of the survey. These faculty members did not have technical knowledge of the subject matter. This approach was adopted to focus attention on the language and structural clarity of the survey since many national counterparts who would be completing the survey speak English as a second language. In the second pilot phase, 9 technical experts on immunization financing and procurement completed an online pilot survey using MachForm, an electronic system for survey and data collection. These experts included public officials from the MoHs of Hong Kong Special Administrative Region of the People's Republic of China (SAR), Macau SAR, China, and Cambodia and spoke English as a second language.

### Sampling frame and survey distribution

WPRO's Expanded Program on Immunization Unit shared contact details of MoH immunization focal points from the 27 countries in the WPR. These MoH focal points were invited to nominate NITAG and MoF counterparts with whom they liaise in vaccine procurement activities. On March 20, 2019, the online MachForm survey was distributed by email to these MoH focal points. Email reminders were sent on day 10 and day 22 after the initial survey was disseminated to encourage completion of surveys.

## Results

### Characteristics of respondents and response rate

The analysis was undertaken from a country perspective. Out of the 80 national stakeholders from 27 countries who were invited to participate in the survey, 17 (21%) stakeholders, representing 13 (48%) countries in the WPR provided responses (S2 Table and S1 Data). Following the World Bank's 2016 income classifications, this subset of 13 countries was made up of 6 high-income countries (HICs), 3 UMICs, and 4 LMICs, of which 2 LMICs were eligible to receive support from Gavi and 2 were transitioning from Gavi support. Five out of the 13 countries were Pacific island countries, while the remaining are part of the wider WPR.

Surveys were completed by 14 current and one former MoH staff and two NITAG members. Due to the low response rate from NITAG members and no responses from MoF staff, only the 15 MoH responses were included in the analysis. Respondents were immunization managers (n = 6) or persons who had another role within the immunization divisions of MoHs (n = 9). Twelve respondents had more than 5 years of work experience in vaccine procurement within their national ministries and institutions. Two countries had two MoH representatives and answers were compared for consistency. Answers pertaining to national procurement processes were consistent among these MoH representatives from the same country while their perspectives on the establishment of a regional pooled procurement process differed slightly. Respondent's identities were unknown unless they agreed to provide contact information and survey results were aggregated to protect the anonymity of survey respondents.

### Country-specific procurement process

**Vaccine forecasting requirements and national approval timelines.** Most respondent countries (11/13) used tools to forecast vaccine requirements annually and reported having both a financial and programmatic multi-year plans; two countries had only one or the other (Table 1). With regard to the frequency with which national vaccine requirements were reviewed, eight countries had annual reviews and three had quarterly reviews; 9 countries (69%) had budget line items specifically for vaccines. Respondents from six countries specified having a timeline of less than 6 months from development of vaccine forecasts to signing of contracts; three said their countries had 6-month timelines; and two reported their countries had one-year timelines. The average time duration between signing contracts with manufacturers and/or suppliers until vaccines were delivered was reported for 10 countries; the process took 3 months or less for six countries, 6 months for three countries and one year for one country.

**National tender and bidding processes.** Five countries had tender and bidding processes that were compulsory for the vaccine's inclusion; eight countries did not have compulsory processes, which included two countries that specified that such processes were not necessary since they were Gavi-eligible. Respondents from five countries reported 1–3 bids as the

**Table 1. Summary of responses related to country-specific procurement processes in the World Health Organization (WHO) Western Pacific Region.**

| Survey Questions | Responses from WHO Western Pacific Region countries[1] (N = 13) Agree (Green), Disagree (Red), Unknown (Yellow) | | |
|---|---|---|---|
| Use of tools to quantify/forecast vaccine requirements annually | 11 | 1 | 1 |
| There are technical specification(s) limiting vaccine qualifications for contract awards | 6 | 4 | 3 |
| Annual national budget line item specifically for vaccines | 9 | 2 | 2 |
| Country has a financial and/or programmatic multi-year long term plan for NIP | 11 | | 2 |
| Payment to vaccine suppliers are typically/always on time | 6 | 2 | 5 |
| Country has a NITAG | 7 | 5 | 1 |
| MoF or Cabinet approves funding if new vaccine is recommended by NITAG/MoH | 10 | 1 | 2 |
| Tender/bidding process is necessary to determine vaccine prices to be included in NIP | 5 | 3 | 5 |
| Every year the same supplier and/or manufacturer typically wins procurement bids | 3 | 5 | 5 |
| Bidding process is repeated in the event that the minimum requirements is not met | 2 | 5 | 6 |
| Regulation(s)/law(s) favoring the selection of nationally manufactured vaccines | 3 | 7 | 3 |
| Country has an NRA | 7 | 3 | 3 |
| Vaccines within NIP are licensed by NRA | 7 | | 6 |
| Vaccine licensure is initiated by vaccine manufacturer(s) or importer | 6 | | 7 |
| Vaccine licensure is initiated by the Government | 1 | | 12 |
| Vaccine pricing has confidentiality agreements with manufacturers | 4 | 3 | 6 |
| In 2017, 100% of vaccines and syringes used in the EPI were covered by the government | 8 | 2 | 3 |
| Country produces some or all vaccines included in countries' NIP | 3 | 10 | |
| Different quality standards for locally produced and imported vaccines | 1 | 1 | 11 |
| Recent challenge/bottleneck affecting national vaccine procurement process | 1 | 7 | 5 |
| Value-added tax or other taxes are imposed on imported and/or locally procured vaccines | 6 | 2 | 5 |
| There were reported vaccine stock outs for the period of 2015–2017[2] | 5 | 6 | 2 |
| There delayed vaccine deliveries during the 2017 contract year[3] | 4 | 4 | 5 |
| Awareness and use of the WHO V3P database | 7(8)[4] | 5 | 1 (0)[4] |
| Country purchases all routine vaccines at a fair price | 7 | 4 | 2 |
| Country purchases all routine vaccines at an affordable price | 9 | 3 | 1 |

EPI–Expanded Program on Immunization

MoF–Ministry of Finance

MoH–Ministry of Health

NIP–National Immunization Program

NITAG–National Immunization Technical Advisory Group

NRA–National Regulatory Authority

V3P –Vaccine, Product, Price Procurement

[1] Number of countries represented by respondents' answers that are in agreement, in disagreement, or are unsure of summary statement

[2] Identified stock-out include: Bacille Calmette-Guerin vaccine (BCG), hepatitis B vaccine (HepB), oral polio vaccine (OPV), diphtheria, tetanus, pertussis-containing vaccines (DTP-CV) and human papillomavirus (HPV) vaccine.

[3] Identified frequency of deliveries ranged from 1–3 times within the contract year.

[4] Country with two surveyed MoH counterparts had inconsistent responses reflecting awareness of V3P database, one confirmed their knowledge while the other was not aware.

minimum number needed for each tender while one country specified 6 as the minimum. Only three countries reported having particular suppliers or manufacturers that consistently won bids every year and 5 countries were unsure if their countries had such a trend.

**Timeliness of national immunization programme financing & taxation.** Eight (62%) countries had NIPs that were fully funded by the government. Respondents from six countries confirmed that payments to suppliers were usually or always timely, those from two countries said they were not able to pay suppliers on time, and those from five countries were unsure if there were payment delays. A respondent from only one country specified that such delay was caused by their country's local vaccine production. Despite having payment delays, vaccine deliveries were not delayed as a result. On the other hand, delivery delays from other causes were reported for four countries. Six (46%) countries had a value-added tax or other taxes imposed on imported and locally procured vaccines; four had taxes above 6% of the vaccine's value and one had taxes ranging from 4–6%.

**Vaccine specifications, licensing & national regulatory authorities.** Six (46%) countries had technical specifications that limited the vaccines that were qualified to receive contract awards. Such technical specifications were based on the following considerations: vaccine presentation, WHO pre-qualification, in-country vaccine registration, and approval granted by National Regulatory Authorities (NRAs). Seven (54%) countries had an NRA that licensed all routine vaccines included in their NIPs.

**Implementation of national immunization technical advisory group recommendations.** Of the seven countries that reported having NITAGs, five always or usually followed NITAG recommendations on new vaccine introductions; of the remaining other two countries with NITAGs, one followed recommendations sometimes and the other did not specify the frequency. Either the government's cabinet or the MoF of eight countries had to approve funding for new vaccines that NITAGs or MoHs had recommended; only one country had another unique national institution that approved such funding.

**Vaccine supply and stock-out frequency.** Respondents from two countries identified vaccine stock-outs and supply issues as the main bottlenecks in their countries' vaccine procurement processes. From 2015–2017, five countries had vaccine stock-outs, which included the Bacille Calmette-Guerin vaccine (BCG), hepatitis B vaccine (HepB), oral polio vaccine (OPV), diphtheria, tetanus, pertussis-containing vaccines (DTP-CV) and human papillomavirus (HPV) vaccine. Respondents from three countries reported the frequency of stock-outs ranged from 1–3 times during the 2017 contract year.

## Vaccine prices and procurement mechanisms

**Perception on vaccine price fairness and affordability.** Respondents from seven (54%) countries thought that their countries procured all routine vaccines (e.g. primary vaccines routinely included and universally offered through NIPs) at fair and affordable prices (Table 1). Respondents from five countries thought that vaccine prices were either unfairly priced or unaffordable, and identified inactivated polio vaccine (IPV), HPV vaccine, measles and rubella (MR) vaccine, pneumococcal conjugate vaccine (PCV), and the single formulation hexavalent vaccine (diphtheria, tetanus, acellular pertussis [DTaP]-HepB-IPV-Haemophilus influenzae [Hib]) as being too expensive for their national immunization budget. Vaccine pricing was perceived as fair by respondents when they compared the price differences among suppliers during the competitive bidding process, but when they compared vaccine prices to those offered by the UNICEF/SD, two respondents changed their perceptions and pricing was viewed as unfair.

**Knowledge on vaccine pricing resource and reporting on vaccine pricing.** Respondents from about half of the countries (7/13) were aware of and/or used the WHO Vaccine Product, Price, and Procurement database that allows individuals to access vaccine pricing data for different procurement mechanisms, income classifications, and WHO regions. Four countries had confidentiality agreements with manufacturers, three did not and respondents from six countries were unsure if such agreements existed. This could have contributed to the poor reporting completeness for the questions related to vaccine pricing. The respondent from only one country reported that paediatric influenza vaccine was offered to all children through their country's NIP, but no vaccine pricing information was provided. Of the 13 routine vaccines listed, DTP-CV was the vaccine for which prices in countries were most commonly disclosed (7/13) (S3 Table). Among countries that reported their vaccine price, the percent difference was lowest for monovalent Hib vaccine (reported for 3 countries) vaccine and highest for DTP-CV (reporting for 7 countries).

**Vaccine price reporting by income classification.** Most countries that reported vaccine pricing were HICs. Of the four LMICs, only one reported vaccine prices for all 7 routine vaccines included in their NIPs and, similarly, only one of the UMICs reported vaccine prices for the 6 routine vaccines included in their NIP. Four of the six HICs had complete reporting of prices for all routine vaccines included in their NIPs except for the paediatric influenza vaccine. PCV and HPV vaccines had the largest price differences among the vaccine prices reported, having price differences of US$137 and US$177, respectively. Prices reported by some UMICs for BCG, HepB, HPV vaccine, DTP-CV and measles vaccines were greater than the lowest prices reported by some HICs. The price of BCG supplied to a UMIC by an independent manufacturer was almost 4 times higher than the lowest price reported by an HIC that procured vaccines through UNICEF's VII (separate from Gavi). The measles, mumps and rubella vaccine (MMR) price reported by one UMIC was 1.15 times greater than the lowest price reported by a HIC (procurement mechanism unspecified). The price of nine-valent HPV vaccine paid by one HIC was less than half of what was paid for bivalent HPV vaccine by one UMIC. Depending on the formulation of the DTP-CV, prices offered to UMICs were 4 to 87 times greater than the lowest HIC-reported price.

**Vaccine procurement mechanism used.** Approximately 40% (27/67) of the reported purchases of 10 routine vaccines were from international vaccine manufacturers (S3 Table). The second most common (30%, 20/67) reported purchases were through the UNICEF/SD (these were for Gavi-ineligible countries). Only six (BCG, HepB, MCV, OPV, rubella vaccine, and JE vaccine) of the 13 vaccines were reported to have been procured directly from national manufacturers.

**Preferred characteristics of procurement mechanisms.** Preferred characteristics of the procurement mechanisms for vaccines showed that respondents from 11 (85%) countries indicated that flexible participation, i.e. annual pooled rather than compulsory procurement (Table 2). In addition, respondents from 9 (69%) countries preferred payment after receipt of goods. Respondents from three countries thought that a third party should manage their country's vaccine procurement process, from bidding to payment operations, respondents from five countries preferred that all aspects of procurement be managed in-country and five respondents were uncertain about this. Countries did show a clear preference for hedging vaccine prices (i.e. predetermining a fixed price) in local currency or providing payments in foreign currency. Over half (8/13) preferred having a fixed vaccine price, while only one preferred a tiered price and four were unsure of the preferences of their countries.

**Perceptions on pooled procurement.** Respondents most commonly identified budget planning, quality of vaccines procured, timely delivery from manufactures, and cost-savings for NIP vaccines as prime benefits of participating in a regional pooled vaccine procurement

**Table 2. Summary of preferred characteristics for a regional pooled procurement mechanism in the World Health Organization (WHO) Western Pacific Region.**

| Themes of survey questions | Possible answers | Responses from WHO Western Pacific Region Countries (N = 13) | | | | | | | | | | | | |
|---|---|---|---|---|---|---|---|---|---|---|---|---|---|---|
| | | 1 | 2 | 3 | 4 | 5 | 6 | 7 | 8 | 9 | 10 | 11 | 12 | 13 |
| **Suitable administrative fee** (percent of the total vaccine cost) | <1% | | | | 4 | | | | | | | | | |
| | 1–3% | | | | 4 | | | | | | | | | |
| | 4–6% | — | | | | | | | | | | | | |
| | above 6% | — | | | | | | | | | | | | |
| | Unsure/Unknown | | | | | 5 | | | | | | | | |
| **Participation** | Flexible participation[1] | | | | | | | | | | | 11 | | |
| | Fixed participation | — | | | | | | | | | | | | |
| | Unknown/Unsure | | 2 | | | | | | | | | | | |
| **Payment method** | Before receipt of goods | 1(2)[2] | | | | | | | | | | | | |
| | After receipt of goods | | | | | | | | 9(8)[2] | | | | | |
| | Unsure/Unknown | | | 3 | | | | | | | | | | |
| **Currency used for procurement** | Foreign currency[3] | | | | | 5 | | | | | | | | |
| | Local currency[4] | | | | | 5 | | | | | | | | |
| | Unsure/Unknown | | | 3 | | | | | | | | | | |
| **Management of procurement process**[5] | Country management | | | | | 5 | | | | | | | | |
| | Third party management | | | 3 | | | | | | | | | | |
| | Unsure/Unknown | | | | | 5 | | | | | | | | |
| **Vaccine pricing mechanism** | Fixed vaccine pricing | | | | | | | | 8 | | | | | |
| | Tiered vaccine pricing | 1 | | | | | | | | | | | | |
| | Unsure/Unknown | | | | 4 | | | | | | | | | |

[1] Annual vaccine procurement is not compulsory and is reviewed and confirmed on a quarterly basis

[2] Country with two surveyed ministry of health counterparts had inconsistent responses, one selected preferred payment before receipt of goods while the other selected payment after receipt of goods

[3] Refers to US$, Euro, etc., using conversion at the time of scheduled payment

[4] Price hedged (protected) in local currency

[5] Management refers to all aspects of the procurement, from bidding to payment operations

mechanism (Table 3). Only two country respondents thought that few benefits would be gained by participating in a regional pooled procurement mechanism. Barriers for a regional pooled vaccine procurement mechanism included previously established national administrative procurement guidelines and regulations (six countries), and insufficient political support (five countries) and not having appropriate NRA guidelines and regulations. Respondents from five countries expressed an interest in participating in a regional pooled procurement mechanism, three were not interested in participating in a regional pooled vaccine procurement mechanism and five were unsure about participation in such a mechanism. Respondents indicated that positive outcomes from participating in a regional pooled procurement mechanism included decreased financial mismanagement and vaccine stock-outs and more efficient vaccine procurement processes. Respondents from only two countries thought that there would be little benefit in establishing a regional procurement mechanism. Some communicated fear of other countries defaulting on vaccine payments, and how this might negatively affect their participation in and their access to vaccines through the regional pooled procurement mechanism. Additional concerns that were voiced included potential difficulties of a country's ability to manage its immunization-related activities while managing its participation in a pooled procurement mechanism and vaccine quality concerns related to meeting halal requirements. Some respondents said that they would need more information on the cost

**Table 3. Summary of perceptions on having a regional pooled procurement mechanism for vaccines in the World Health Organization (WHO) Western Pacific Region.**

| Themes of survey questions | Possible answers | Responses from WHO Western Pacific Region countries (N = 13) |
|---|---|---|
| | (Respondents could provide more than one answer for the benefits and potential barriers sections, whereas answers specifying interest for participation were mutually exclusive) | Responses Displayed in Descending Order |
| **Benefits or improvements from participation** | Timely vaccine delivery from manufacturer* | 10 |
| | Budget planning for vaccines | 9 |
| | Quality of vaccines procured | 9 |
| | Cost-saving on NIP/EPI vaccines | 9 |
| | Payment after the receipt of goods | 8 |
| | Transparency of procurement process* | 8 |
| | Financial oversight of funds allocated to immunization* | 6 |
| | Reduced administrative costs* | 6 |
| | None or very little benefits/improvement* | 2 |
| | Unsure/Unknown | 1 |
| **Potential barriers affecting participation** | Inadequate/No National Administrative Procurement guidelines and regulations | 6 |
| | Insufficient political will & support | 5 |
| | Inadequate/No National Regulatory Authority guidelines and regulations | 5 |
| | Irregular and/or unreliable government funding for vaccines* | 3 |
| | Unsure/Unknown | 3 |
| | National vaccine manufacturer | 1 |
| **Interested in participation** | Yes | 5 |
| | Unsure/Unknown | 5 |
| | No | 3 |

* Answer only reported by one of the two surveyed counterparts for the two countries with more than one response

EPI–Expanded Program on Immunization

NIP–National Immunization Program

of vaccines, the quality of vaccines and delivery times by manufacturers before determining whether it would be advantageous for their countries to participate in a regional pooled procurement mechanism.

## Discussion

Respondents from 5 of the 13 countries that completed the survey expressed interest in participating in a regional pooled procurement mechanism, 5 indicated they needed more information on the pooled procurement mechanism or would prefer to defer responses to a higher authority, and three were not in favour of participating in a regional pooled procurement mechanism. Vaccine pricing disparities for five routine vaccines were notable between UMICs and Pacific island HICs that procured through the UNICEF VII, with the Pacific island HICs having access to lower vaccine prices than the UMICs that procured vaccines through independent suppliers/manufacturers. Respondents from three countries reported having unreliable vaccine funding which would affect their countries' successful participation in a regional procurement mechanism. If these countries were to default on vaccine payments, this could affect the functionality of the whole regional pooled procurement mechanism. Such a concern was identified as a potential drawback that would affect interest in participating in a regional pooled procurement mechanism. For its successful establishment, countries would have to

introduce or update national regulations and legislation. Institutionalizing a country's participation in a regional pooled procurement mechanism would ensure that sufficient funds are secured and allocated for immunization. As seen in the AMR from 2000–2013, the 29 countries that have passed immunization-related legislation spent 98% more on immunization for having ensured secured funding and institutionalized oversight over the program's financing than those that do not have immunization-related provisions [3]. This has allowed countries to secure funds for immunization, protecting countries from defaulting on payments to the RF which has maintained its long-standing success and function.

Responses pertaining to vaccine supply, procurement processes, vaccine prices, financing, and quality assurance obtained from the 15-country survey conducted in 2003 by EURO overlap with those collected in our WPR survey [12]. Similar to our findings, one-third of those surveyed in EURO identified unreliable vaccine supplies as the cause for delays in vaccine procurement and shipping, leading to vaccine shortages. Transparency in national vaccine procurement processes was a concern among the surveyed European countries as well as for more than half of surveyed countries in the WPR. Unlike the European counterparts, vaccine pricing was not reported to be a bottleneck for vaccine implementation in most countries in the WPR, although cost-savings was identified as a potential benefit from participating in a regional pooled procurement mechanism by almost three-quarters of WPR respondents. Respondents from both regions perceived that a pooled procurement mechanism would improve availability and access to quality vaccines from different manufacturers. High vaccine prices and substantial differences in prices of vaccines procured by countries in different income categories were also observed in both regions. Lack of appropriate guidelines and regulations from NRAs and the challenges in assuring quality vaccines were also reported by those surveyed in both regions. Despite the geographic differences and the 15-year time gap between the two surveys, similar themes arose in responses from both regions. Future studies in the WPR should follow a similar methodology as used by DeRoeck et al., such as using in-depth face-to-face interviews with selected national stakeholders, to build upon the current study's findings.

There were several limitations that may have affected the results of this study. Having representation by just under half (13/27) of WPR countries and completion of surveys by only 21% of potential respondents makes it difficult to conclude that responses fully represent the region's perceptions and interest in participating in a regional pooled procurement mechanism. However, there was a fairly balanced distribution of countries with respect to population size and location in WPR, with 31% (4/13) of countries being highly populated countries and 38% (5/13) being Pacific island countries. The 13 countries participating in the survey represent 22% of the region's population and would be representative of 71% of the WPR if China, which represents 69% of the region's population, were to be excluded. Three countries out of the top five most populated countries that make up 93% of the region's surviving infant population are represented in the survey. The initial intent was to obtain perspectives from MoF and NITAG counterparts, but there were no MoF responses and the two NTIAG were excluded, so the findings only reflect MoH perspectives. Responses provided by each country's representative may not be reflective of the overall perceptions of the MOH and other decision makers. Furthermore, the number of "unsure" responses represented only 7% of the survey results obtained, which indicates that most questions were generally understood by respondents. On the other hand, these "unsure" answers could be a reflection that either the individual who was invited to participate in the survey was not involved or aware of their country's procurement process, that questions were not sufficiently clear to those surveyed, or that respondents did not wish to provide information on vaccine pricing and procurement because of other concerns. Results could have also been influenced by response bias, i.e. those supporting pooled procurement mechanisms being more or less likely to respond. Answers provided

could likely reflect the opinions of those surveyed rather than being representative of their country's interest in participating in a regional pooled procurement mechanism. Some respondents communicated that they felt it was not appropriate to complete a survey on the behalf of an entire ministry. Self-selection bias might therefore be present in the results of this study since national counterparts had a choice as to whether or not they would like to participate in completing the survey. We were not able to assess the degree to which MoH respondents and non-respondents were similar or different and so could not assess the degree of response bias. National stakeholders might have been inclined to provide answers aligned with global and regional strategies related to pooled procurement as the topic appears in conferences and meetings, reflecting social desirability bias. There was also the possibility that the email survey did not successfully reach all national stakeholders and there might have been hesitation to complete an online survey.

Budget planning, quality of vaccines, timely delivery, cost-savings and payment after receipt were benefits identified by the respondents from most participating countries. Insufficient political will and support, not having adequate NRA guidelines, and national administrative procurement regulations were identified as potential barriers to establishing a regional pooled procurement mechanism by almost half of the country respondents. Participants from more than half of the participating countries preferred having fixed vaccine prices and only one selected a tiered pricing mechanism, while the remaining four were uncertain about this issue. While there is some uncertainty, respondents from 5 of the 13 countries supported a regional pooled vaccine procurement mechanism whereas respondents from only 3 countries were hesitant to support such a mechanism, specifying the need for more information for decision-making and approval to participate in such mechanism from a higher authority; however, respondent from 5 countries did not respond to this question. Despite the 16-year gap between surveys, the perceived benefits of establishing a pooled procurement mechanism among WPR respondents were similar to those of EURO respondents. It is imperative that mechanisms that increase vaccine affordability and access be explored in the WPR. Future research exploring the feasibility of establishing a regional pooled procurement mechanism are critical for informing the regional strategy towards reaching the objectives of the new Global Immunization Agenda 2030 and of the Regional Strategic Framework for Vaccine Preventable Disease and Immunization 2021–2030 in the Western Pacific to improve the financial sustainability of NIPs and improve the transparency of vaccine pricing in the region in this new decade.

## Supporting information

**S1 Table. Comparison of themes within vaccine pooled procurement surveys.**
(PDF)

**S2 Table. Summary of respondents to survey assessing perceptions of establishing a vaccine pooled procurement mechanism for Western Pacific Region.**
(PDF)

**S3 Table. Summary of vaccine prices and procurement mechanisms as reported in survey assessing perceptions of establishing a vaccine pooled procurement mechanism for Western Pacific Region.**
(PDF)

**S1 Data. Anonymized dataset.**
(XLS)

## Acknowledgments

The authors would like to thank all national stakeholder from the Western Pacific Region who participated in this survey, all those participated in the piloting of the survey and Denise DeRoeck and Peter Carrasco for their technical feedback.

## Author Contributions

**Conceptualization:** Alice Abou-Nader, E. Anthony S. Nelson.

**Data curation:** Alice Abou-Nader.

**Formal analysis:** Alice Abou-Nader.

**Investigation:** Alice Abou-Nader.

**Methodology:** Alice Abou-Nader, James D. Heffelfinger, Ananda Amarasinghe, E. Anthony S. Nelson.

**Project administration:** Alice Abou-Nader.

**Supervision:** James D. Heffelfinger, E. Anthony S. Nelson.

**Writing – original draft:** Alice Abou-Nader.

**Writing – review & editing:** James D. Heffelfinger, Ananda Amarasinghe, E. Anthony S. Nelson.

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
