## [Decision Letter · Decision Letter 0]

10 May 2022

PGPH-D-22-00163

Assessing perceptions of establishing a vaccine pooled procurement mechanism for the Western Pacific Region

Dear Dr. Nelson,

Thank you for submitting your manuscript to PLOS Global Public Health. After careful consideration, we feel that it has merit but does not fully meet PLOS Global Public Health’s publication criteria as it currently stands. Therefore, we invite you to submit a revised version of the manuscript that addresses the points raised during the review process.

It has been hard to recruit reviewers lately, but I do not want to hold up the publication of your paper, especially with one very positive review in hand.  Please consider the suggestions they have made, and also these points that come from my own reaction when reading your paper:

Country-specific procurement process (Line 149 - 191) - it could be a bit overwhelming to have all of these results together in a single paragraph. Consider restructuring thematically, into smaller paragraphs, grouping results from related questions together.

Same comment for Vaccine Prices and Procurement Mechanisms Section (Line 196 - 239)

Supplementary Table 3, and in particular, the range in vaccine prices that you found seem worthy of more prominent reporting, perhaps even in a figure.  Am I reading correctly that the reported Hep B vaccine price varies 10,000%, from $0.20 to $20.88?  That is just striking.

Please submit your revised manuscript by . If you will need more time than this to complete your revisions, please reply to this message or contact the journal office at globalpubhealth@plos.org. Please include the following items when submitting your revised manuscript:

We look forward to receiving your revised manuscript.

Kind regards,

Abraham D. Flaxman, Ph.D.

Academic Editor

Journal Requirements:

1. Please provide additional details regarding participant consent. In the ethics statement in the Methods and online submission information, please ensure that you have specified whether consent was written or verbal/oral. If consent was verbal/oral, please specify: 1) whether the ethics committee approved the verbal/oral consent procedure, 2) why written consent could not be obtained, and 3) how verbal/oral consent was recorded. If your study included minors, please state whether you obtained consent from parents or guardians in these cases. If the need for consent was waived by the ethics committee, please include this information.

2. Please provide an Author Summary. This should appear in your manuscript between the Abstract (if applicable) and the Introduction, and should be 150–200 words long. The aim should be to make your findings accessible to a wide audience that includes both scientists and non-scientists. Sample summaries can be found on our website under Submission Guidelines: 

https://journals.plos.org/globalpublichealth/s/submission-guidelines#loc-parts-of-a-submission

3. Please amend your detailed Financial Disclosure statement. This is published with the article. It must therefore be completed in full sentences and contain the exact wording you wish to be published.

4. If you have no competing interests to declare, please state "The authors have declared that no competing interests exist". Otherwise please declare all competing interests beginning with the statement "I have read the journal's policy and the authors of this manuscript have the following competing interests:" Kindly, update the Competing Interest section in the system.

5. We have noticed that you have uploaded Supporting Information files, but you have not included a list of legends. Please add a full list of legends for your Supporting Information files after the references list. 

6. In the online submission form, you indicated that "If the manuscript is accepted for publication arrangements will be made to make the data available through the Chinese University of Hong Kong's data respository system". All PLOS journals now require all data underlying the findings described in their manuscript to be freely available to other researchers, either 1. In a public repository, 2. Within the manuscript itself, or 3. Uploaded as supplementary information.

Additional Editor Comments (if provided):

Reviewers' comments:

Reviewer's Responses to Questions

**Comments to the Author**

1. Does this manuscript meet PLOS Global Public Health’s publication criteria? Is the manuscript technically sound, and do the data support the conclusions? The manuscript must describe methodologically and ethically rigorous research with conclusions that are appropriately drawn based on the data presented.

Reviewer #1: Yes

2. Has the statistical analysis been performed appropriately and rigorously?

Reviewer #1: Yes

3. Have the authors made all data underlying the findings in their manuscript fully available (please refer to the Data Availability Statement at the start of the manuscript PDF file)?

Reviewer #1: Yes

4. Is the manuscript presented in an intelligible fashion and written in standard English?

Reviewer #1: Yes

5. Review Comments to the Author

Reviewer #1: Good and clear article. It is a pity not more counties responded; yet it gives a good impression of feelings towards pooled vaccine procurement.

I have two suggestions for the tables 1 and 3, as follows:

The table 1 is clear and yet not very easy to grasp at a glance. I would suggest that the numbers in the three columns be replaced by a horizontal stacked bar-chart (with numbers printed in the bars), which would make it immediately clear which characteristics are generally found in large numbers of countries – especially if colour codes are used. I would then present the bar-sections as yes (green) undecided (yellow) and no (red).

Table 3: I would re-order them, starting always with the statement of highest agreement (e.g. timely vaccine delivery, 10 points)

6. PLOS authors have the option to publish the peer review history of their article (what does this mean?). If published, this will include your full peer review and any attached files.

**Do you want your identity to be public for this peer review?** For information about this choice, including consent withdrawal, please see our Privacy Policy.

Reviewer #1: **Yes: **Prof Dr Hans V Hogerzeil

---

## [Decision Letter · Decision Letter 1]

28 Jun 2022

Assessing perceptions of establishing a vaccine pooled procurement mechanism for the Western Pacific Region

PGPH-D-22-00163R1

Dear Prof Nelson,

We are pleased to inform you that your manuscript 'Assessing perceptions of establishing a vaccine pooled procurement mechanism for the Western Pacific Region' has been provisionally accepted for publication in PLOS Global Public Health.

Best regards,

Abraham D. Flaxman, Ph.D.

Academic Editor

Reviewer Comments (if any, and for reference):

Reviewer's Responses to Questions

**Comments to the Author**

1. If the authors have adequately addressed your comments raised in a previous round of review and you feel that this manuscript is now acceptable for publication, you may indicate that here to bypass the “Comments to the Author” section, enter your conflict of interest statement in the “Confidential to Editor” section, and submit your "Accept" recommendation.

Reviewer #1: All comments have been addressed

2. Does this manuscript meet PLOS Global Public Health’s publication criteria? Is the manuscript technically sound, and do the data support the conclusions? The manuscript must describe methodologically and ethically rigorous research with conclusions that are appropriately drawn based on the data presented.

Reviewer #1: Yes

3. Has the statistical analysis been performed appropriately and rigorously?

Reviewer #1: Yes

4. Have the authors made all data underlying the findings in their manuscript fully available (please refer to the Data Availability Statement at the start of the manuscript PDF file)?

Reviewer #1: Yes

5. Is the manuscript presented in an intelligible fashion and written in standard English?

Reviewer #1: Yes

6. Review Comments to the Author

Reviewer #1: Thank you for changing the bar chart and fig 3.

7. PLOS authors have the option to publish the peer review history of their article (what does this mean?). If published, this will include your full peer review and any attached files.

**Do you want your identity to be public for this peer review?** For information about this choice, including consent withdrawal, please see our Privacy Policy.

Reviewer #1: **Yes: **Prof Dr Hans V Hogerzeil, University Medical Centre Groningen, Netherlands
